# Beyond the Last Answer: Your Reasoning Trace Uncovers More than You Think

## Abstract

Large Language Models (LLMs) leverage step-by-step reasoning to solve complex problems. Standard evaluation practice involves generating a complete reasoning trace and assessing the correctness of the final answer presented at its conclusion. In this paper, we challenge the reliance on the final answer by posing the following two questions: Does the final answer reliably represent the model's optimal conclusion? Can alternative reasoning paths yield different results? To answer these questions, we analyze intermediate reasoning steps, termed *subthoughts*, and propose a method based on our findings. Our approach involves segmenting a reasoning trace into sequential subthoughts based on linguistic cues. We start by prompting the model to generate continuations from the end-point of *each* intermediate subthought. We extract a potential answer from every completed continuation originating from different subthoughts. We find that aggregating these answers by selecting the most frequent one (the mode) often yields significantly higher accuracy compared to relying solely on the answer derived from the original complete trace. Analyzing the consistency among the answers derived from different subthoughts reveals characteristics that correlate with the model's confidence and correctness, suggesting potential for identifying less reliable answers. Our experiments across various LLMs and challenging mathematical reasoning datasets (AIME2024 and AIME2025) show consistent accuracy improvements, with gains reaching up to 13% and 10% respectively.

## 1 Introduction

Large Language Models (LLMs) have demonstrated remarkable capabilities in solving complex tasks when prompted to articulate their reasoning process step-by-step (Wei et al., 2022). Reasoning requires not only a sufficiently rich knowledge base acquired during pre-training but also increased computational resources during inference (test-time compute). This allows models to engage in a deliberate, multi-step reasoning process akin to human "System 2 thinking" (Kahneman, 2011), moving beyond immediate, intuitive "System 1" responses (Kahneman, 2011). Models like OpenAI's o1 (Jaech et al., 2024) and DeepSeek-R1 (Guo et al., 2025) attest to the importance of scaling test-time compute by dedicating substantial inference resources to generate elaborate reasoning traces before producing a final output. Standard evaluation protocols for such models typically focus exclusively on the final output, i.e., the model generates a reasoning trace culminating in a final answer, and only this *single final answer* is evaluated for correctness.

However, relying on the final answer potentially overlooks valuable information encoded within the reasoning process itself. It implicitly assumes that the single generated path represents the model's definitive reasoning, neglecting the possibility that slight variations in the thought process could lead to different, and perhaps more accurate, conclusions. This raises a fundamental question: ***Can we establish a more reliable assessment of an LLM's reasoning ability by analyzing the evolution and consistency of its answers throughout the reasoning process?***

In this paper, we propose a method to investigate this question by probing the internal consistency of an LLM's reasoning. Our core idea involves interrupting the reasoning process at intermediate points, or

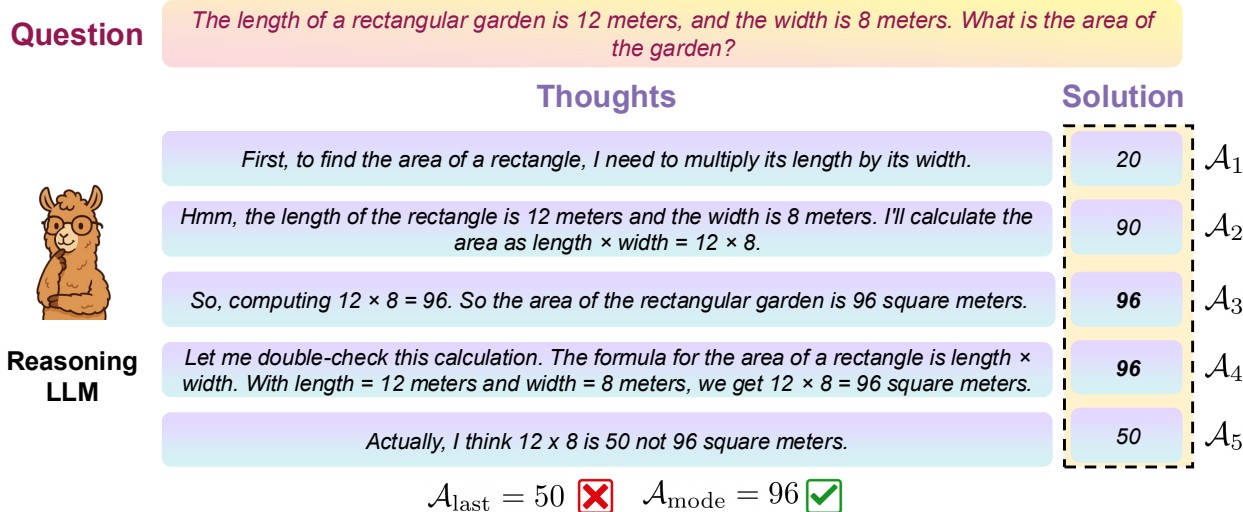

Figure 1: **Subthought Analysis.** We show that by examining intermediate reasoning steps and their corresponding answers $(A_1, \ldots, A_5)$, taking the mode of these answers $(A_{mode})$ often leads to better performance than using only the final answer $(A_{last})$, as is typically done. This figure illustrates a case where $A_{mode} = 96$ is correct, while $A_{last} = 50$ is not.

"subthoughts", and examining the conclusions reached from these states as illustrated in Figure 1. Specifically, our methodology entails:

1. Generating an initial, complete reasoning trace for a given problem using standard greedy decoding.
2. Segmenting this trace into a sequence of subthoughts based on natural linguistic markers that often indicate shifts or progressions in reasoning (e.g., "Wait," "Alternatively," "Hmm").
3. Prompting the *same* model to generate a complete solution starting from an intermediate state (i.e., after each cumulative sequence of subthoughts).
4. Extracting the final numerical answer derived from *each* of these generated continuations, thereby producing a set of potential answers that reflect conclusions reached from various points within the initial reasoning structure.

This process yields a distribution of answers for the original problem. We analyze this distribution with two primary goals: First, we investigate how the model's answer evolves across different subthought stages. We examine whether the final answer in the original trace is consistently reached from earlier points. We also look into how the distribution of answers differs between problems the model ultimately answers correctly versus incorrectly. We hypothesize that inconsistent or high variability in the answers across different subthought sequences might indicate difficulty or potential errors, serving as a signal of low confidence or hallucination.

Second, based on the insights from this analysis, we explore whether aggregating the collected answers can lead to a more robust final result. Specifically, we hypothesize that the most frequently occurring answer (the mode) across all generated completions represents a more reliable conclusion, reflecting convergence across slightly perturbed reasoning trajectories.

Our experiments on challenging mathematical reasoning datasets (AIME2024, AIME2025) using seven open-weight LLMs validate these hypotheses. We observe that the consistency patterns indeed differ for correctly and incorrectly solved problems. Furthermore, aggregating answers via the mode significantly improves accuracy compared to using only the final answer from the initial trace, demonstrating the practical benefit of our analysis.

**Our contributions are:**

- A methodology for systematically analyzing LLM reasoning by generating and evaluating conclusions derived from intermediate subthoughts.

- An analysis showing how answer consistency evolves during the reasoning process, revealing distinct patterns for correct versus incorrect solutions and suggesting potential for error detection based on answer distribution characteristics (e.g., entropy).
- Empirical evidence demonstrating that aggregating answers from subthought completions, specifically by taking the mode, significantly improves accuracy over the standard final-answer approach (up to 13% on AIME2024, 10% on AIME2025).
- A comparison of greedy versus non-greedy sampling strategies for generating subthought completions, highlighting their respective advantages.

We believe these findings offer valuable insights into the nature of LLM reasoning and suggest that evaluating beyond the single final answer can unlock a more nuanced understanding of model capabilities and lead to improved performance evaluation and potentially new reasoning strategies. We proceed by detailing our methodology, followed by the experimental setup and results, and conclude with a discussion of implications.

## 2 Related Work

**Test-Time Scaling and Reasoning.** Chain-of-thought (CoT) prompting (Wei et al., 2022) is a pivotal technique for scaling test-time or inference-time compute. It explicitly asks an LLM to generate a structured reasoning chain before arriving at the final answer. Self-consistency CoT (Wang et al., 2023b) is a CoT variant that replaces greedy decoding with sampling-based decoding to sample multiple reasoning chains and select the best answer through consistency aggregation. Other prompting techniques focus on constructing reasoning-provoking structured prompts (Paranjape et al., 2021; Sanh et al., 2021; Mishra et al., 2021). Search-and planning-based prompting techniques divide the reasoning task into a set of sub-tasks (Dua et al., 2022; Zhou et al., 2022; Khot et al., 2022; Suzgun & Kalai, 2024). These methods can be categorized into approaches that evaluate the final outcome or the reasoning process (Lightman et al., 2023). Prompting-based test-time scaling techniques guide the model to select the best reasoning chain without updating its parameters. Our approach utilizes the full reasoning chain generated from vanilla CoT prompting. It then prompts the same model with a growing sequence of subthoughts to elicit an answer at different reasoning-trace lengths. The resulting answer distribution is then aggregated by taking the mode, akin to self-consistency CoT. It is worth noting that our method can be utilized with any test-time scaling method that generates an explicit series of subthoughts.

**Training-Based Reasoning.** Training-based techniques train the model to enhance its reasoning capabilities. The key challenge for these methods is the scarcity of human-annotated step-by-step reasoning chains. Research in this direction focuses on developing techniques to automatically generate valid reasoning traces or proposing training methods that effectively leverage the available data. The most straightforward approach for training reasoning models is to finetune a model with supervised finetuning (SFT) on reasoning trajectories (Huang et al., 2024; Min et al., 2024). Other works have shown that preference learning further improves reasoning capabilities. (Min et al., 2024; Hui et al., 2024; Jiao et al., 2024) all explore DPO Rafailov et al. (2023). (Zhang et al., 2024; Lai et al., 2024) explore step-level DPO instead of outcome-level DPO. More recent methods bypass the need for annotated reasoning chains by leveraging reinforcement learning (RL). A particular success in this direction is GRPO Shao et al. (2024), which shows that RL is sufficient for the emergence of complex reasoning capabilities even without an initial supervised fine-tuning step. The methods discussed so far use explicit natural language reasoning traces. A recent line of work explores latent reasoning methods that represent reasoning chains implicitly. These methods focus on compressing natural language chains into a much smaller number of tokens Deng et al. (2023; 2024). Other works introduce learnable tokens that are intended to enable the model to perform additional non-verbal steps before outputting an answer token (Goyal et al., 2023; Wang et al., 2023a). More effectively, Hao et al. (2024) proposed using the last-layer hidden feature as implicit reasoning tokens that are fed back to the model to generate the next token autoregressively. Our method is a test-time method and does not update model parameters. It works with any reasoning model that outputs an explicit natural-language thought process before the final answer.

**Overthinking Phenomenon in Reasoning Models.** The overthinking phenomenon in reasoning models occurs when the model generates excessively detailed and redundant reasoning steps for relatively simple problems (Chen et al., 2024). This phenomenon compromises the inference efficiency of reasoning models and

in some cases leads to incorrect answers. Several recent works explicitly address computational efficiency and reasoning quality by imposing a length-based reward to control the length of CoT reasoning Arora & Zanette (2025); Yeo et al. (2025). The s1 approach Muennighoff et al. (2025) introduced "budget forcing" to effectively control compute through targeted prompt modifications. Similarly, L1 Aggarwal & Welleck (2025) introduced Length Controlled Policy Optimization (LCPO), precisely managing reasoning complexity. Contrary to budgeted reasoning techniques, our method operates in a high-compute regime.

Our method is inspired by the observation that overthinking may lead to wrong answers. It analyzes the dynamics of the thought process as the model generates longer reasoning traces. It extracts a self-consistent answer and provides insight into correctness by measuring the entropy of the model's answers.

## 3 Methodology

We introduce a framework for analyzing LLM reasoning by examining the conclusions derived from intermediate steps ("subthoughts") within an initial reasoning trace. The process involves: 1) generating an initial trace, 2) segmenting it based on linguistic cues, 3) prompting completions from these intermediate points, 4) extracting the resulting answers, and 5) analyzing the distribution of these answers.

### 3.1 Problem Setting and Initial Trace Generation

Let $P$ represent a problem statement that requires complex reasoning (e.g., a mathematical proof or calculation). We employ a reasoning language model, denoted by $\mathcal{M}$, to solve $P$. The process begins by formulating an initial prompt, $\Pi(P)$, designed to instruct $\mathcal{M}$ to provide step-by-step reasoning enclosed within specific delimiters (e.g., `<thought>...</thought>`) followed by a final answer in a designated format (e.g., `\boxed{Answer}`).

Using this prompt, we generate an initial full response $R_{full}$ via greedy decoding to obtain the model's most probable reasoning path:

$$R_{full} = \mathcal{M}(\Pi(P), \text{Params}_{greedy})$$

From this full response $R_{full}$, we extract two critical components:

- The primary reasoning trace $T$, typically identified as the content within the final `<thought>...</thought>` block.
- The final answer $A_{last}$, extracted from the concluding part of $R_{full}$, usually conforming to the `\boxed{...}` format. This extraction is performed by a dedicated extraction procedure or model, denoted $\mathcal{M}_{extract}$.

The answer $A_{last}$ serves as the baseline for comparison, corresponding to the standard approach of taking the single answer produced at the end of the initial trace.

### 3.2 Subthought Identification and Segmentation

At the core of our method is segmenting the initial reasoning trace $T$ into a sequence of meaningful intermediate steps or *subthoughts*, denoted $(s_1, s_2, \ldots, s_n)$. This segmentation aims to capture points where the model might pause, reflect, change direction, or move to a distinct next step in its reasoning.

We perform segmentation based on occurrences of words or phrases from a predefined set $W$, which we refer to as **Subthought Transition Markers**. These markers often signal reflection, correction, sequencing, or the exploration of alternatives. The set $W$ used in our experiments is:

**Subthought Transition Markers** *(W)*

```
"Wait", "Alternatively", "Another angle", "Another approach", "But wait", "Hold on",
  "Hmm", "Maybe", "Looking back", "Okay", "Let me", "First", "Then", "Alright", "Got it",
    "I don't see any errors", "I think", "Let me double-check", "Let's see", "Now",
  "Remember", "Seems solid", "Similarly", "So", "Starting", "That's correct", "That seems
                        right", "Therefore", "Thus"
```

We utilize regular expressions derived from $W$ to split the trace $T$. The pattern ensures that a transition marker from $W$ typically indicates the start of a new subthought chunk $s_j$ (for $j > 1$), and the marker itself is included at the beginning of $s_j$. If no markers from $W$ are found within $T$, the entire trace is treated as a single subthought ($n = 1$). Letting $\oplus$ denote string concatenation, the original trace can be reconstructed as $T = s_1 \oplus s_2 \oplus \cdots \oplus s_n$.

### 3.3 Subthought Completion Generation

For each identified subthought boundary $i \in \{1, 2, \ldots, n\}$, we construct a cumulative partial thought trace $T_i$, representing the reasoning up to the end of subthought $s_i$:

$$T_i = s_1 \oplus s_2 \oplus \cdots \oplus s_i$$

We then create a modified prompt $P_i$ based on the original prompt $\Pi(P)$. This prompt $P_i$ contains the original problem description but replaces the full reasoning trace $T$ with the partial trace $T_i$. $P_i$ is formatted such that $T_i$ appears within the appropriate reasoning delimiters (e.g., `<thought>...</thought>`) and ends in a way that prompts the model $\mathcal{M}$ to continue the reasoning process from that specific state. Let $\text{Format}(\Pi(P), T_i)$ represent this formatting function:

$$P_i = \text{Format}(\Pi(P), T_i)$$

Each partial prompt $P_i$ is then fed back into the *same* reasoning model $\mathcal{M}$ to generate a completion $C_i$. The concatenation $R_i = P_i \oplus C_i$ forms a complete response initiated from the intermediate state $T_i$. This process is repeated for each $i$ from 1 to $n$, resulting in $n$ full response variations.

We experiment with two distinct sampling strategies for generating these completions $C_i$:

- **Greedy Subthought Completion** ($\text{Params}_{greedy}$): Uses temperature = 0.0 and top-p = 1.0. This strategy forces the model to follow its deterministic, highest-probability reasoning path continuation from the state defined by $T_i$.
- **Non-Greedy Subthought Completion** ($\text{Params}_{diverse}$): Uses temperature = 1.0 and top-p = 0.95. This encourages stochasticity and allows the model to explore alternative, potentially less probable but still viable, reasoning paths extending from $T_i$.

It is important to note that the initial trace $T$ (used for segmentation) is always generated using $\text{Params}_{greedy}$. The choice between greedy and non-greedy strategies applies only during the generation of the $n$ completions $C_i$ from the partial prompts $P_i$.

### 3.4 Answer Extraction from Completions

For each of the $n$ generated response variations $R_i = P_i \oplus C_i$, corresponding to completions starting after subthoughts $s_1, \ldots, s_n$, we extract the final numerical answer $A_i$. This extraction uses the same procedure or model $\mathcal{M}_{extract}$ employed for obtaining $A_{last}$:

$$A_i = \mathcal{M}_{extract}(R_i)$$

$\mathcal{M}_{extract}$ is designed to robustly identify and isolate the final numerical answer, parsing specific formats like `\boxed{...}` or identifying the most salient numerical result if the expected format is absent. This procedure yields a set of $n$ potential answers for the original problem $P$:

$$\mathcal{A} = \{A_1, A_2, \ldots, A_n\}$$

This set $\mathcal{A}$ captures the conclusions reached by the model when forced to complete the reasoning process from different intermediate stages.

### 3.5 Analysis and Aggregation Metrics

The set of answers $\mathcal{A}$ forms the basis for our analysis. As detailed in the results section, we first analyze the properties of this set, such as the evolution of answers $(A_1, \ldots, A_n)$ and their distribution (e.g., using consistency measures or entropy) to understand how stability relates to correctness.

Based on this analysis, we evaluate the effectiveness of aggregating these answers. Let $A_{true}$ be the ground truth answer for problem $P$. We define an indicator function for correctness: $\text{IsCorrect}(A, A_{true}) = 1$ if answer $A$ matches $A_{true}$, and 0 otherwise. We compare performance using two primary metrics, averaged over a dataset of problems:

1. **Last Answer Accuracy ($\text{Acc}_{Last}$):** The accuracy of the single answer $A_{last}$ extracted from the initial, uninterrupted greedy trace $R_{full}$. This serves as our baseline.

$$\text{Acc}_{Last} = \mathbb{E}_{P \sim \text{Dataset}}[\text{IsCorrect}(A_{last}, A_{true})]$$

2. **Most Frequent Answer Accuracy ($\text{Acc}_{MostFreq}$):** The accuracy of the most frequent answer (mode) $A_{mode}$ within the set $\mathcal{A} = \{A_1, \ldots, A_n\}$ derived from subthought completions.

$$A_{mode} = \underset{A \in \mathcal{A}}{\text{argmax}} \left( \sum_{j=1}^{n} \mathbb{I}(A_j = A) \right)$$

where $\mathbb{I}(\cdot)$ is the indicator function. Ties for the mode are broken by selecting the answer that appeared earliest in the sequence $(A_1, \ldots, A_n)$ (i.e., the one corresponding to the smallest index $j$).

$$\text{Acc}_{MostFreq} = \mathbb{E}_{P \sim \text{Dataset}}[\text{IsCorrect}(A_{mode}, A_{true})]$$

Our central hypothesis, explored in the experiments, is that analyzing the set $\mathcal{A}$ provides valuable insights. Specifically, aggregating the responses by mode yields a significant improvement over the baseline in both greedy and non-greedy completion strategies $\text{Acc}_{MostFreq} \geq \text{Acc}_{Last}$.

## 4 Experiments

### 4.1 Experimental Setup.

**Datasets.** We consider two datasets, AIME2024 and AIME2025, derived from the American Invitational Mathematics Examination. Both datasets are known to require substantial reasoning capabilities.

**Models.** In order to evaluate our hypothesis, we consider seven open source models: DeepScaleR-1.5B-Preview, DeepSeek-R1-Distill-Qwen-1.5B, DeepSeek-R1-Distill-Qwen-14B, EXAONE-Deep-7.8B, Light-R1-7B-DS, QwQ-32B, and Skywork-OR1-Math-7B. For answer extraction ($\mathcal{M}_{extract}$), we consistently used `Qwen/Qwen2.5-14B-Instruct`, prompted specifically to extract the final numerical answer in the `\boxed{...}` format or identify the most likely and final numerical result otherwise.

**Implementation Details.** For efficient inference, we build our pipeline on the vLLM library Kwon et al. (2023). We limit the maximum number of newly generated tokens at every LLM call to 8192.

### 4.2 Analysis of Answer Evolution Across Subthoughts

We first investigate how the final answer $A_i$ derived from completing the reasoning after the $i$-th subthought ($T_i$) evolves as $i$ increases from 1 to $n$. Figure 2 illustrates this evolution for different models on selected problems from AIME2024 using greedy subthought completion. Each plot shows the sequence of answers $A_1, \ldots, A_n$ (y-axis) against the subthought index $i$ (x-axis, labeled "Candidate Index"). The plots also mark

the ground-truth answer ($A_{true}$), the final answer from the original full trace ($A_{last}$), and the most frequent answer from the sequence ($A_{mode}$).

We observe distinct patterns:

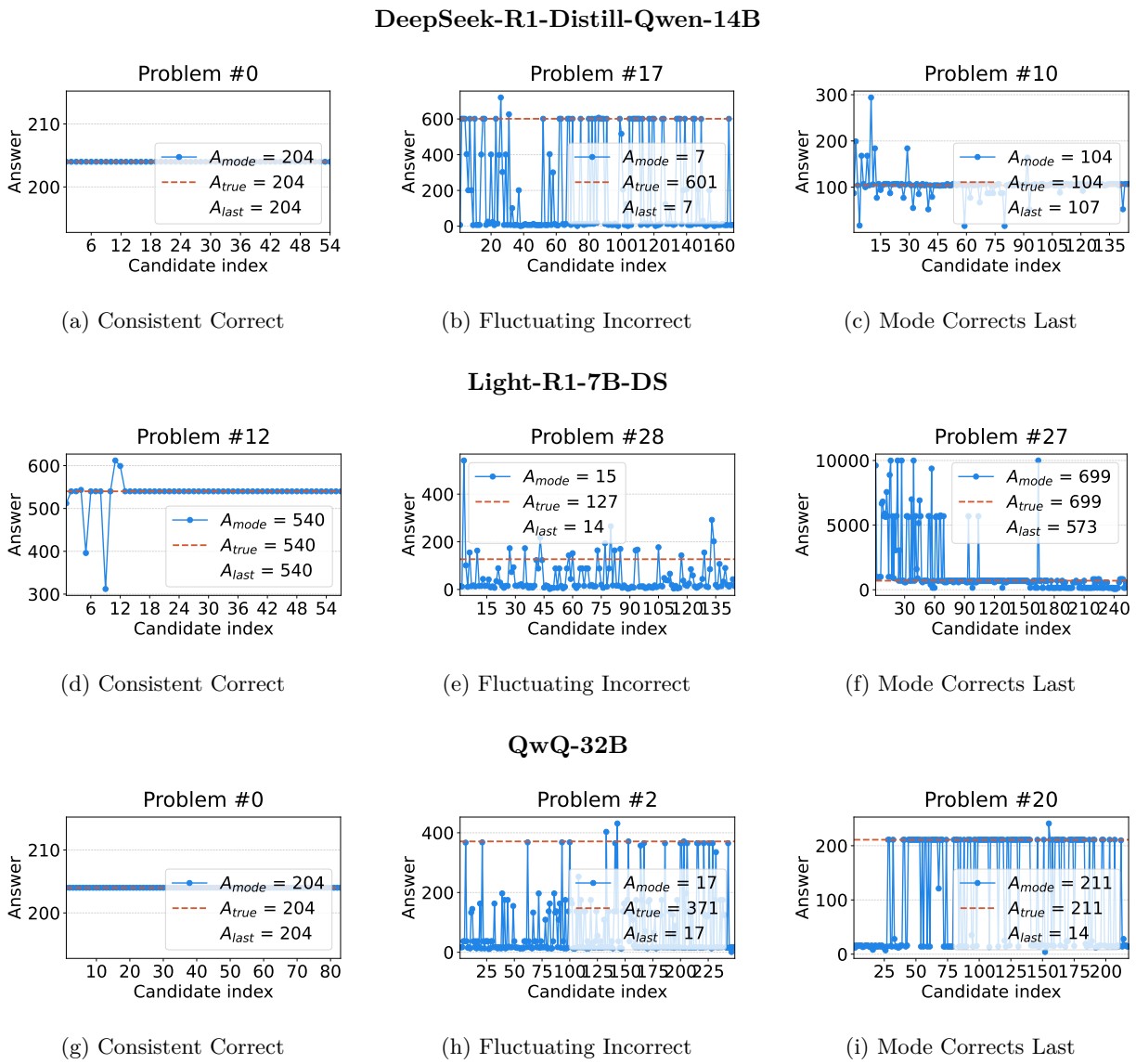

Figure 2: **Answer Evolution Across Models on AIME2024 (Greedy Completion).** Each row corresponds to a different model.

- **Consistent Correctness (e.g., Left):** When the model solves the problem correctly and confidently, the sequence of answers $(A_1, \ldots, A_n)$ often converges quickly to the correct answer $A_{true}$. In these cases, $A_{last} = A_{true}$ and $A_{mode} = A_{true}$. The answers derived from most subthoughts are identical and correct, indicating stable reasoning.
- **Fluctuating Incorrectness (e.g., Middle):** When the model struggles with a problem and produces an incorrect final answer ($A_{last} \neq A_{true}$), the sequence of answers often exhibits high fluctuation. Many different incorrect answers are generated, and the most frequent answer $A_{mode}$ is also typically incorrect. Sometimes, the true answer $A_{true}$ appears sporadically or not at all. This suggests unstable reasoning or exploration of incorrect paths.

- **Mode Corrects Last Answer Error (e.g., Right):** Perhaps the most interesting case is when the initial trace yields an incorrect final answer ($A_{last} \neq A_{true}$), but analyzing the subthought completions reveals a consistent, correct answer ($A_{mode} = A_{true}$). This occurs when the model frequently reaches the correct conclusion from various intermediate states, but the specific path taken in the initial greedy generation happens to derail near the end. This highlights scenarios where $A_{last}$ is misleading, while $A_{mode}$ captures a more robust consensus from the model's internal states.

Note: we did not observe cases in our experiments where $A_{last}$ was correct but $A_{mode}$ was incorrect.

### 4.3 Answer Distribution Entropy and Correctness

The visual patterns in Figure 2 suggest that the distribution of answers in $\mathcal{A} = \{A_1, \ldots, A_n\}$ carries information about the model's reasoning process. To quantify the diversity or uncertainty within this distribution, we calculate the Shannon entropy for each problem:

$$H(\mathcal{A}) = - \sum_{a \in \text{Unique}(\mathcal{A})} p(a) \log_2 p(a)$$

where $p(a) = \frac{1}{n} \sum_{j=1}^{n} \mathbb{I}(A_j = a)$ is the frequency of answer $a$ in the sequence $\mathcal{A}$. Higher entropy indicates a wider spread of different answers (less consistency), while lower entropy indicates convergence towards one or a few answers (more consistency).

Figure 3 compares the average entropy of the answer distributions for problems that were ultimately answered correctly (using $A_{last}$ as the final answer) versus those answered incorrectly, across three different models on AIME2024 with greedy completions.

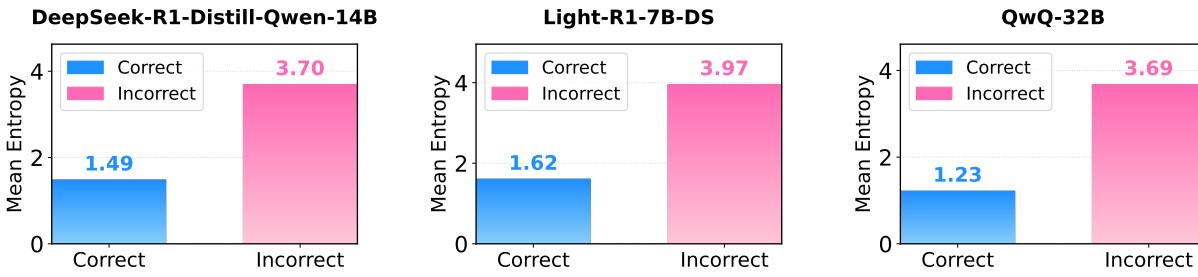

Figure 3: **Mean Entropy of Subthought Answer Distributions on AIME2024 (Greedy Completion).** Comparison between problems answered correctly ($A_{last} = A_{true}$) and incorrectly ($A_{last} \neq A_{true}$). Lower entropy correlates with correct answers.

Across all models, we observe a clear trend: the average entropy for correctly answered problems is significantly lower than for incorrectly answered problems. This quantitatively confirms the visual observation that successful reasoning paths tend to exhibit higher internal consistency across subthoughts, while unsuccessful paths are often characterized by high variability and exploration of diverse (incorrect) conclusions. In future work, this signal could be presented to the user as an indicator of how likely the model's answer is to be correct.

### 4.4 Subthought Aggregation Boosts Accuracy

Building on the qualitative insights, we now quantitatively evaluate our central hypothesis: Does using the most frequent answer ($A_{mode}$) lead to higher accuracy than using the last answer ($A_{last}$)? We compare $\text{Acc}_{MostFreq}$ with the baseline $\text{Acc}_{Last}$ across all tested models and both datasets (AIME2024, AIME2025). We also investigate the impact of the subthought completion strategy by reporting results for both Greedy and Non-Greedy completions.

Figure 4 presents the main accuracy results. The figure compares the baseline Last Answer Accuracy (Acc$_{Last}$, blue bars) with the Most Frequent Answer Accuracy (Acc$_{MostFreq}$, orange bars) under four conditions: AIME2024 with Greedy completions (top-left), AIME2024 with Non-Greedy completions (top-right), AIME2025 with Greedy completions (bottom-left), and AIME2025 with Non-Greedy completions (bottom-right). The numbers above the bars indicate the absolute accuracy percentage point difference (Acc$_{MostFreq}$ − Acc$_{Last}$).

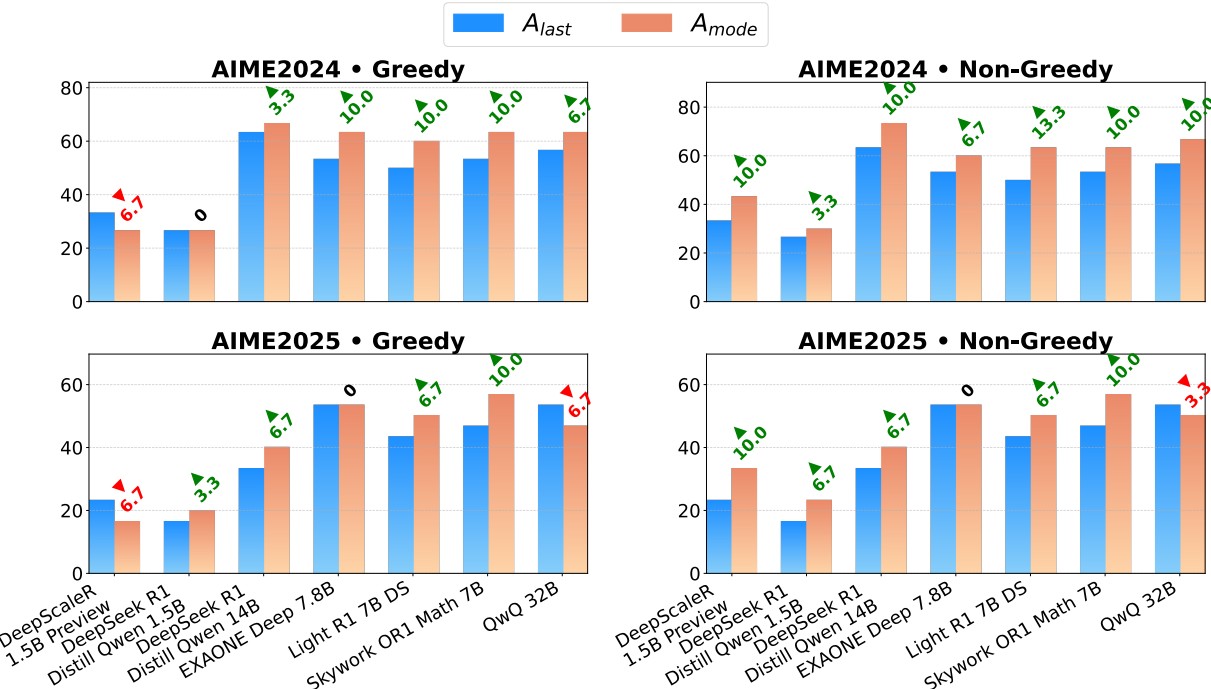

Figure 4: **Accuracy Comparison: Last Answer vs. Most Frequent Answer.** Comparison of Last Answer Accuracy (Acc$_{Last}$, blue) with Most Frequent Answer Accuracy (Acc$_{MostFreq}$, orange) using Greedy and Non-Greedy subthought completions across various models and AIME datasets. Numbers above bars show the absolute gain (Acc$_{MostFreq}$ − Acc$_{Last}$). Green upward triangles indicate improvement, red downward triangles indicate decrease. Our method consistently improves or matches baseline accuracy.

The results strongly support our hypothesis. Aggregating answers using the mode (Acc$_{MostFreq}$) consistently outperforms or matches the baseline accuracy (Acc$_{Last}$) across almost all models, datasets, and completion strategies. The improvements can be substantial:

- On **AIME2024**, gains reach up to **+13.33%** (Light-R1-7B-DS, Non-Greedy) and frequently exceed +6%, with several models showing +10% gains (e.g., DeepSeek-R1-Distill-Qwen-14B, Skywork-OR1-Math-7B, QwQ-32B under Non-Greedy).
- On **AIME2025**, gains reach up to **+10.0%** (DeepSeek-R1-Distill-Qwen-14B Greedy, Skywork-OR1-Math-7B Non-Greedy), with multiple models showing gains over +6%.
- Even when the gain is 0%, our method generally does not hurt performance significantly. The few minor decreases observed (-6.66% for DeepScaleR on AIME2024 Greedy; -3.33% for QwQ-32B on AIME2025 Non-Greedy) might be attributed to noise, particularly for smaller models, or specific problem interactions rather than a systemic flaw.

When comparing the subthought completion strategies, **Non-Greedy** completion tends to yield slightly larger or more frequent improvements than Greedy completion, especially visible on the AIME2024 results (e.g., compare top-left vs. top-right panels for Light-R1, Skywork, QwQ-32B). This suggests that exploring alternative reasoning paths via sampling (Non-Greedy) is often more effective at revealing the model's robust

consensus answer compared to simply reinforcing the most likely path from intermediate states (Greedy). Nonetheless, Greedy completion also provides consistent benefits over the baseline method.

Importantly, the pattern of improvement holds across a diverse set of models (ranging from 1.5B to 32B parameters) and both challenging AIME datasets. This consistency highlights the general applicability and robustness of analyzing subthought stability and using the mode answer as a more reliable indicator of the model's reasoning outcome than the single last answer.

## 5    Conclusion

We demonstrated that evaluating Large Language Models based solely on the final answer of a reasoning trace can be suboptimal. By analyzing intermediate *subthoughts* within a single trace and aggregating the answers derived from completing these partial thoughts, we show the following:

> **Conclusions:**
>
> 1. **Mode Aggregation Enhances Accuracy:** Selecting the most frequent answer ($A_{mode}$) from completions originating at intermediate subthoughts significantly boosts accuracy compared to relying solely on the final answer ($A_{last}$) of the initial trace. Gains of up to +13% on AIME2024 and +10% on AIME2025 are observed across various models.
> 2. **Answer Consistency Signals Reliability:** The distribution of answers generated from subthoughts provides a valuable signal. High consistency (low entropy) correlates strongly with correct baseline solutions ($A_{last}$), while high fluctuation (high entropy) is characteristic of incorrect solutions or model struggle. This suggests potential for using distribution metrics for confidence estimation or error detection.
> 3. **Non-Greedy Completion Often Maximizes Gains:** While both greedy and non-greedy subthought completions improve accuracy via mode aggregation, non-greedy sampling (T=1.0, top-p=0.95) frequently yields larger improvements, likely by better exploring the reasoning space around the initial path segments.

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

## A   Entropy Plots

Plots of the mean entropy of the subthought answer distribution for AIME2024 and AIME2025 are shown in Figure 5 and Figure 6. As observed in Section 4.3, the mean entropy for correctly solved problems is always lower than that of incorrectly solved problems, highlighting that models tend to produce a more diverse set of answers across subthoughts for questions they have a hard time solving correctly.

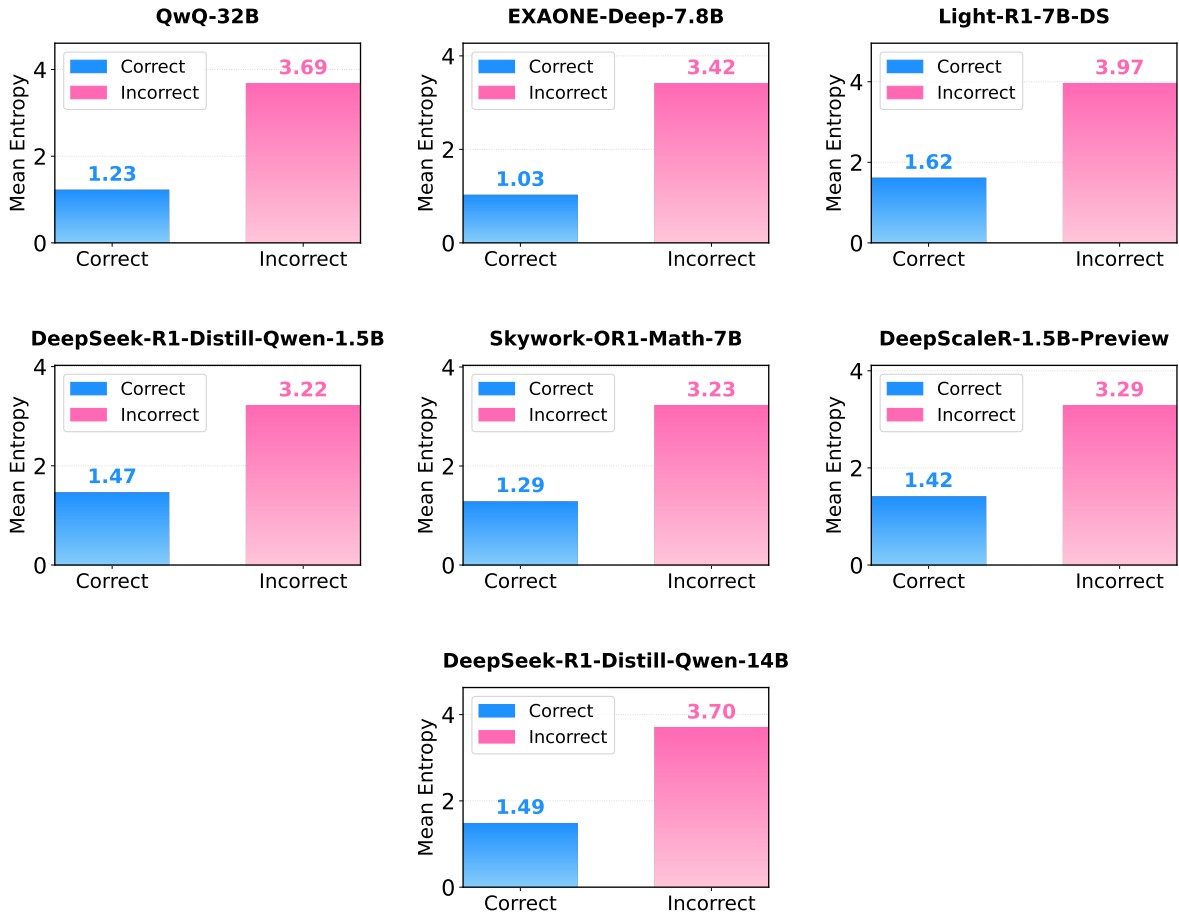

Figure 5: **Mean Entropy of Subthought Answer Distributions on AIME2024 (Greedy Completion).** Comparison between problems answered correctly ($A_{last} = A_{true}$) and incorrectly ($A_{last} \neq A_{true}$). Lower entropy correlates with correct answers.

## B   Use of Language Models in Writing

We used large language models (LLMs) (ChatGPT, Gemini and Claude) to assist in drafting and refining the text of this paper. These tools supported clarity, coherence, and stylistic consistency throughout the writing process.

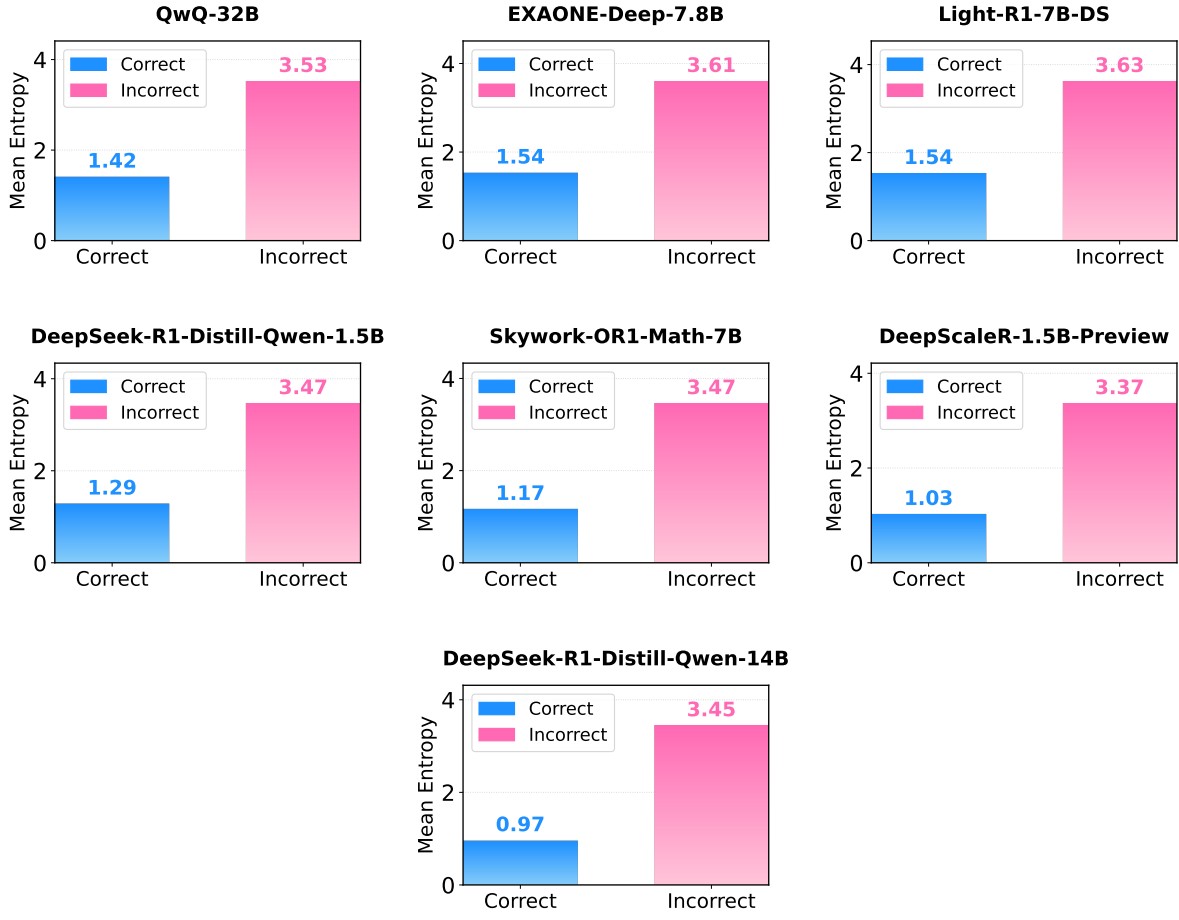

Figure 6: **Mean Entropy of Subthought Answer Distributions on AIME2025 (Greedy Completion).** Comparison between problems answered correctly ($A_{last} = A_{true}$) and incorrectly ($A_{last} \neq A_{true}$). Lower entropy correlates with correct answers.

