# OpenReview forum: "Beyond the Last Answer: Your Reasoning Trace Uncovers More than You Think"
_TMLR — Rejected by TMLR_

### Review · Reviewer_Sgpd · 2026-04-27

**Summary Of Contributions:**

The paper proposes a test-time method that segments an LLM’s reasoning trace into intermediate “subthoughts,” continues generation from each prefix, and aggregates the resulting answers by taking the mode. The authors show on AIME2024/2025 across several open-weight models that this mode answer can outperform the original final answer, suggesting that reasoning traces contain useful information beyond their endpoint. A strength is that the method is simple and directly targets late-stage reasoning derailment or overthinking. A key weakness is that the experiments do not fully separate the proposed subthought mechanism from the benefits of extra sampling/compute, and the results are limited to small numerical-answer math datasets.

**Audience:**

Yes

**Audience Explanation:**

The paper studies a timely issue for TMLR readers: how to evaluate and improve reasoning models beyond simply checking the final answer. Its finding that intermediate reasoning prefixes can sometimes recover a better answer is relevant to work on test-time scaling, chain-of-thought, self-consistency, and reliability estimation. Even if the current evidence is limited, the phenomenon is interesting and may motivate better analyses of reasoning traces and overthinking failures.

**Broader Impact Concerns:**

The main broader impact concern is that the method may encourage stronger reliance on model-generated reasoning traces as reliability signals. If answer consistency or low entropy is interpreted as confidence, users may overtrust confidently repeated but incorrect answers, especially in high-stakes domains. There is also a potential evaluation-gaming issue: if models know that intermediate answers are aggregated, they may learn to repeat or bias candidate answers within the reasoning trace to influence the final mode. A Broader Impact Statement should discuss these risks and clarify that the method is currently evaluated only on short-answer math problems, not safety-critical decision-making settings.

**Claims And Evidence:**

No

**Claims Explanation:**

The paper provides clear evidence that, in the tested AIME settings, aggregating answers from subthought continuations can outperform using the original final answer. However, the evidence is not fully convincing for the broader claims about subthoughts revealing a more reliable reasoning conclusion, because the method is not compared against compute-matched self-consistency, random prefix continuation, or other multi-sampling baselines. The entropy analysis is suggestive, but it only shows a correlation with correctness rather than demonstrating a reliable confidence or error-detection mechanism. Overall, the empirical results support a promising observation, but the current evidence does not fully isolate the proposed mechanism or establish broad generality.

**Requested Changes:**

- Add compute-matched baselines, especially standard self-consistency from scratch, multi-sampling with the same number of generations/tokens, and random or fixed-interval prefix continuation. This is necessary to show that the gains come from the proposed subthought mechanism rather than simply from extra sampling and aggregation.
- Add ablations for the subthought segmentation strategy. The paper should compare the proposed marker-based segmentation against sentence-level splitting, fixed-token splitting, random splitting, and different marker sets, to establish that the linguistic “subthought” boundaries are meaningful.
- Report statistical uncertainty. Since AIME2024/2025 are small datasets, the paper should include raw correct counts, confidence intervals, and paired significance tests where appropriate.
- Clarify and moderate the claims about confidence/error detection. The entropy results are currently correlational; stronger evidence such as AUROC, coverage-accuracy curves, or calibration analysis would be needed to support reliability-estimation claims.

---

### Review · Reviewer_K9CA · 2026-05-04

**Summary Of Contributions:**

This paper proposes a method to guide the decision process of an LLM toward more accurate answers in the context of mathematical exercises. The approach consists of splitting the original flow of thought into multiple sub-thoughts using hardcoded transition markers. Multiple candidate answers are then generated by taking the mode of completions drawn from variants of the original textual thought process. The authors show that the method outperforms the original unguided answer. Additionally, they propose leveraging the process to produce uncertainty estimates by design and to analyze patterns in failures and successes.

Strengths:

- The writing is clear and easy to follow.
- The method is straightforward and leads to substantial improvements within the scope of the experiments.
- The use of the process to quantify uncertainty is a clever and appealing idea.

Weaknesses:

- The variety of datasets used is very limited: only AIME 2024 and AIME 2025, which are quite similar to each other.
- The method is only compared against raw answers, with no comparison to other existing test-time intervention methods. This is particularly surprising given that the authors appear to be aware of concurrent methods, as discussed in the related work section.
While the analysis subsections are welcome, they consist of short discussions of anecdotal examples rather than theoretically or statistically grounded insights.
- There is no discussion of the computational cost of the method, despite the process appearing to be expensive.
- The method relies on hardcoded transition markers, which may cause robustness issues.

**Additional Comments:**

[1] Wang, Xuezhi, et al. "Self-Consistency Improves Chain of Thought Reasoning in Language Models." The Eleventh International Conference on Learning Representations.

[2] Chen, Qiguang, et al. "Towards reasoning era: A survey of long chain-of-thought for reasoning large language models." arXiv preprint arXiv:2503.09567 (2025).

**Audience:**

No

**Audience Explanation:**

Taken in isolation, the findings could be of interest. However, the manuscript fails to situate itself adequately within the existing literature. While the authors do highlight related work, there is a lack of insight into how the proposed method performs relative to and differs from prior approaches. This risks misleading the reader into thinking the test-time intervention literature is narrower than it actually is.

**Claims And Evidence:**

No

**Claims Explanation:**

The weaknesses of the experimental evaluation are, in my view, the main shortcoming of this paper. There is a lack of variety in both datasets and metrics (only accuracy on two very similar datasets), and the analysis section relies primarily on anecdotal examples.

**Requested Changes:**

Major:

- Include a broader range of datasets in the experiments.
- Include comparisons with other intervention methods, or provide a clear argument for why they are not comparable to the proposed framework. As a concrete example, self-consistency [1] is the most obvious concurrent method. Furthermore, the proposed solution appears closely related to what the survey in [2] describes as parallel scaling.

Minor:

- Add a table reporting computational costs. Ideally, compare with alternatives of similar computational budget.
- Broaden the analyses toward less anecdotal and more statistically grounded insights.

---

### Review · Reviewer_XB7z · 2026-05-09

**Summary Of Contributions:**

The paper investigates whether the standard practice of evaluating an LLM's reasoning solely by the final answer of a complete reasoning trace is reliable. The authors propose a Subthought Analysis framework that (1) generates a full greedy reasoning trace, (2) segments it into a sequence of subthoughts, (3) prompts the same model to generate a fresh completion starting from each cumulative partial trace, (4) extracts a final answer A_i from each completion, and (5) aggregates the results by mode or via the Shannon entropy of its distribution. Greedy and non-greedy sampling strategies for completion are compared.

On AIME2024 and AIME2025 with seven open-weight reasoning-tuned models, the authors observe strong accuracy gains from the mode answer over the original last answer.  On correctly-solved problems the mean answer-distribution entropy is much lower. The authors also provide qualitative case studies of three behavioural patterns: "Consistent Correct", "Fluctuating Incorrect", "Mode Corrects Last." No case of "correct original last answer but incorrect mode answer" was observed.

Strengths:
- The paper is generally well written and includes sufficient detail for the reader to reproduce the experiments.
- The "Mode Corrects Last" phenomenon is an interesting observation and challenges the current evaluation practice.
- The entropy contrast between correct and incorrect problems is robust across all models and datasets, and is therefore a convincing empirical result.

Weaknesses:
- The results in the paper generally lacks statistical significance justifications. The accuracy claims rely on 30-problem datasets without confidence intervals reported.
- The headline accuracy result lacks the self-consistency (Wang et al., 2023b) baseline at matched compute, Without it, the paper cannot claim that subthought structure drives the gains, rather than simple sample diversification.

**Audience:**

Yes

**Audience Explanation:**

LLM for math reasoning is a very popular research topic and evaluating math reasoning tasks in LLMs with the final answer has been the standard practice in the field. This paper gives a number of interesting observations on this topic. In particular, the framing of "the final answer is not always the model's best answer" is a useful challenge to current evaluation practice and may inform future work on better aggregation, verification, or self-consistency variants for reasoning models.

**Broader Impact Concerns:**

I have no significant ethical concerns specific to this work. The research operates at inference time on standard mathematical reasoning benchmarks.

**Claims And Evidence:**

No

**Claims Explanation:**

- For core claims on accuracy improvements, such as the claim that the accuracy of the mode consistently improves over the accuracy of the last, the authors did not provide any analysis on statistical significance. The analysis in the paper was done with 30 problems per dataset and single-seed runs. No confidence intervals or statistical tests were provided. While the numbers does look somewhat convincing, I would like the authors to be a bit more rigorous here with the results.
- The claim that "non-Greedy completion often maximizes gains" is also based on only once sample from each $T_i$. Control experiments on greedy with multiple repeats per T_i, or Non-Greedy applied to the full prompt (i.e., self-consistency), is needed before this can be attributed to subthought completion.

**Requested Changes:**

Critical:
- Provide statistical evidence for the accuracy gains. Re-state the magnitudes ("up to 13%") in light of the resulting uncertainty.
- Add a random-truncation ablation. Replace the marker-based segmentation with random truncation at the same number of token positions (or at random subthought-length-matched positions) and report the resulting AccMostFreq and entropy. This isolates whether the subthought structure drives the effects, rather than any prefix perturbation.
- Disentangle gains from non-greedy sampling. Either (a) sample multiple completions per T_i so that prefix-variance and sampling-variance can be separated, or (b) include a non-greedy self-consistency baseline on the full prompt, so the reader can see how much of the non-greedy advantage is attributable to subthought conditioning.

Would strengthen the work:
- Add a self-consistency baseline at matched compute.
- Discuss the compute cost explicitly. Report total decoded tokens for the subthought pipeline relative to a single greedy trace, and acknowledge that the method is presently prohibitive for online inference.

---

### Comment · Action_Editor_KE8w · 2026-05-19
**Author Discussion Period**

Hello Authors,

This is a reminder that we are currently in the Author–Reviewer discussion phase for this paper. Please use this period to respond to the reviewers’ comments and/or revise the manuscript as needed.

---

### Decision · Action_Editor_KE8w · 2026-06-14

**Recommendation:** Reject

**Additional Comments:**

Reviewers raised their concerns in the reviews, but the authors did not respond during the rebuttal period. Two out of three have recommended rejection. The third reviewer's recommendation is missing, but as their concerns stay unaddressed, I would recommend rejecting the work.

Authors are recommended to substantiate the claims with more experiments before resubmitting.

**Audience:**

Yes

**Audience Explanation:**

The topic is relevant.

**Claims And Evidence:**

No

**Claims Explanation:**

As highlighted by the reviewers, the claims in the paper require more thorough evaluation on diverse datasets.